# High-throughput expansion microscopy enables scalable super-resolution imaging

John H Day[1], Catherine M Della Santina[1†], Pema Maretich[2†], Alexander L Auld[2†], Kirsten K Schnieder[2], Tay Shin[3], Edward S Boyden[1,3,4,5,6,7,8,9], Laurie A Boyer[1,2,9]*

[1]Department of Biological Engineering, Massachusetts Institute of Technology, Cambridge, United States; [2]Department of Biology, Massachusetts Institute of Technology, Cambridge, United States; [3]Department of Media Arts and Sciences, Massachusetts Institute of Technology, Cambridge, United States; [4]Department of Brain and Cognitive Sciences, Massachusetts Institute of Technology, Cambridge, United States; [5]McGovern Institut, Massachusetts Institute of Technology, Cambridge, United States; [6]Howard Hughes Medical Institute, Massachusetts Institute of Technology, Cambridge, United States; [7]K Lisa Yang Center for Bionics, Massachusetts Institute of Technology, Cambridge, United States; [8]Center for Neurobiological Engineering, Massachusetts Institute of Technology, Cambridge, United States; [9]Koch Institute, Massachusetts Institute of Technology, Cambridge, United States

*For correspondence: lboyer@mit.edu

[†]These authors contributed equally to this work

## eLife Assessment

This **important** study develops a high throughput version of expansion microscopy that can be performed in 96-well plates. The engineered technology is **convincing** and compatible with standard microplates and automated microscopes and thus will be of broad interest. The application to hiPCS-derived cardiomyocytes treated with doxorubicin provides a **solid** proof-of-concept demonstrating the potential for high-throughput analysis.

**Abstract** Expansion microscopy (ExM) enables nanoscale imaging using a standard confocal microscope through the physical, isotropic expansion of fixed immunolabeled specimens. ExM is widely employed to image proteins, nucleic acids, and lipid membranes in single cells; however, current methods limit the number of samples that can be processed simultaneously. We developed High-throughput Expansion Microscopy (HiExM), a robust platform that enables expansion microscopy of cells cultured in a standard 96-well plate. Our method enables ~4.2 x expansion of cells within individual wells, across multiple wells, and between plates. We also demonstrate that HiExM can be combined with high-throughput confocal imaging platforms to greatly improve the ease and scalability of image acquisition. As an example, we analyzed the effects of doxorubicin, a known cardiotoxic agent, on human cardiomyocytes (CMs) as measured by the Hoechst signal across the nucleus. We show a dose-dependent effect on nuclear DNA that is not observed in unexpanded CMs, suggesting that HiExM improves the detection of cellular phenotypes in response to drug treatment. Our method broadens the application of ExM as a tool for scalable super-resolution imaging in biological research applications.

## Introduction

The ability to resolve subcellular features can greatly enhance biological discovery, identification of disease targets, and the development of targeted therapeutics. Super-resolution microscopy methods such as structured illumination microscopy (SIM), stochastic optical reconstruction microscopy (STORM), and stimulated emission depletion microscopy (STED) enable nanoscale imaging of specimens (*Gustafsson, 2000*; *Hell and Wichmann, 1994*; *Rust et al., 2006*). However, these approaches require specialized expertise, costly reagents and microscopes. ExM is an inexpensive and accessible method that similarly resolves fluorescently labeled structures at subcellular resolution by isotropically enlarging specimens within a swellable polyelectrolyte hydrogel (*Chen et al., 2015*). Images of expanded cells show a higher effective resolution due to an increase in the distance between the fluorescent molecules. Super-resolution imaging and ExM are critical tools that are widely used in biological research; however, sample processing and imaging require time-consuming manual manipulation, limiting throughput. Although there has been success in increasing the throughput of certain super-resolution microscopy techniques, these methods require highly specialized equipment and expertise which limits the accessibility of these tools (*Holden et al., 2014*; *Gunkel et al., 2014*; *Mahecic et al., 2020*; *Xie et al., 2023*). To circumvent these limitations, we developed HiExM that combines parallel sample processing and automated high-content confocal imaging in a standard 96-well cell culture plate.

## Results

Adapting standard ExM to a 96-well plate requires the reproducible delivery of a small volume of gel solution to each well (<1 μL compared to ~200 μL/sample for slide-based ExM). This small volume is necessary to allow complete gel expansion within the size constraints of wells in a 96-well plate. To achieve this goal, we engineered a device that retains small liquid droplets when dipped into a reservoir of monomeric expansion gel solution. These droplets can then be deposited into the centers of wells across the cell culture plate (*Figure 1a and b*). The retention of the gel solution is mediated by a series of reticulated grooves at the end of a cylindrical post which allows for the formation of a pendant droplet (*Figure 1c*). We estimate that each post retains ~230 nL of gel solution as measured by the mass of liquid accumulated on all post tips of a single device after dipping (Supplementary methods).

The small volume of gel solution delivered to each well by the device forms a toroidal droplet where the inner surface is molded by the conical surface of the post-tip and the outer surface is constrained by surface tension (*Figure 1c and d*). The post-tip dimensions allow delivery of a volume of gel solution that expands relative to the diameter of the well (*Figure 1d*). The toroidal gel enables isotropic gel expansion within the well constraints while allowing for easy device removal because the inner surface of the gel expands outward away from the conical post-tip, effectively detaching the gel from the device. To achieve this toroidal geometry, our device design incorporates three key features to ensure each post of the device contacts the midpoint at the bottom of each well (*Figure 1—figure supplement 1*). First, two sets of three cantilevers ('pressure struts') are located at each end of the device and flank two rows of six posts (12 posts per device). These pressure struts apply outward force to the inside of the outer four wells ensuring that the device remains in contact with the well surface throughout the expansion process. Second, the spine of the device incorporates several notches extending through the long side of the device allowing it to bend to accommodate plate inconsistencies between the depths of individual wells. Finally, the post lengths are offset from the center four posts to allow for sequential deformation that ensures each post contacts the well surface. Together, these design features allow for reproducible deposition and expansion of gels in a 96-well plate.

To test our device for high-throughput gel expansion, we first performed ProExM using ammonium persulfate (APS) and tetramethylethylenediamine (TEMED) to initiate gel polymerization (*Figure 1—figure supplement 2*; *Tillberg et al., 2016*). Since APS rapidly accelerates the polymerization of polyacrylamide hydrogels, TEMED-containing expansion gel solution is first delivered by the device and removed leaving a ≤1 μl droplet in each well of the plate on ice. Another device is then used to deliver APS-containing expansion gel solution at the same volume and the polymerization reaction is initiated by placing the plate on a 50 °C surface. Given the large air-liquid-interface of the gel, droplet delivery and polymerization are performed in a nitrogen-filled glove bag to minimize oxygen inhibition of the

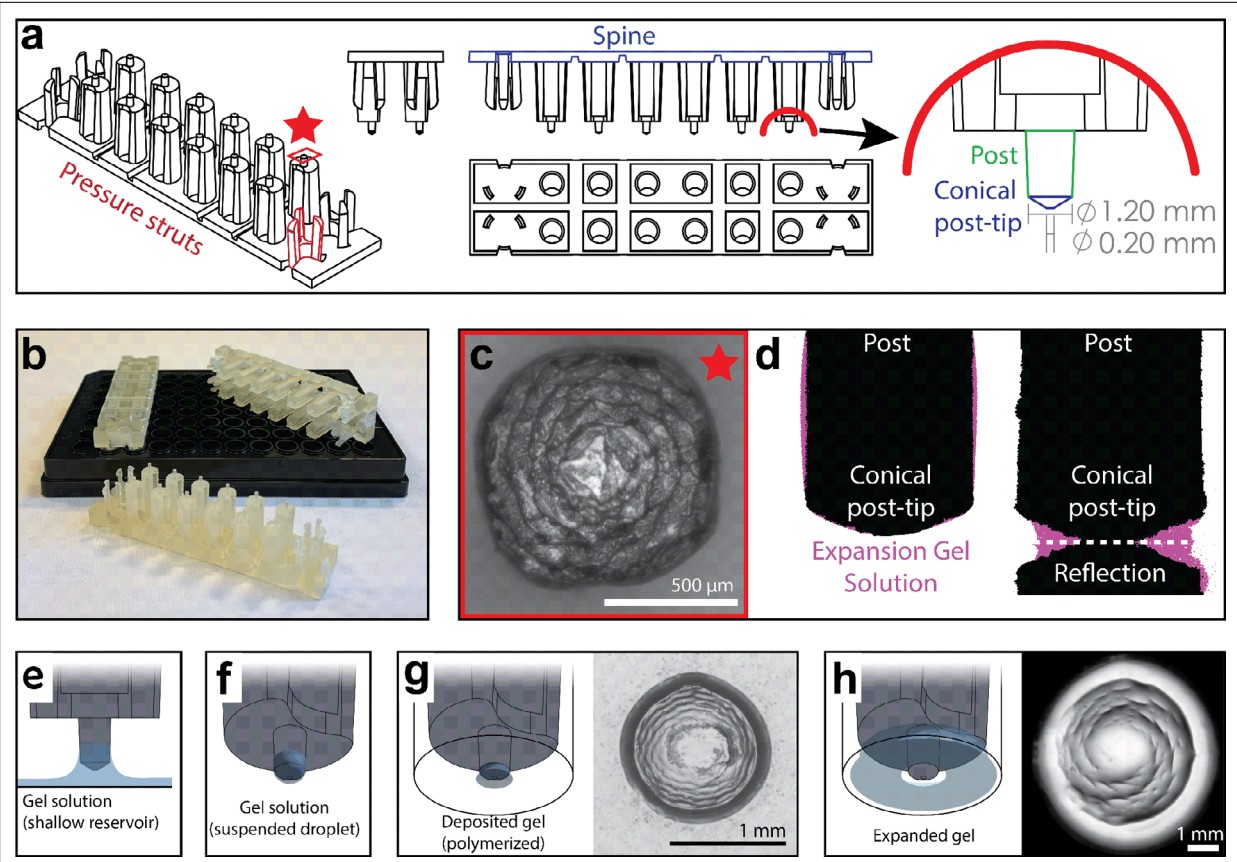

**Figure 1.** High-throughput expansion microscopy (HiExM) enables gel formation and expansion in a 96-well cell culture plate. (**a**) Schematic representation of HiExM devices showing the key features highlighted in color. (**b**) Example devices used in 96-well cell culture plates. (**c**) Brightfield image of the conical post-tip shows the pattern of grooves that mediate fluid retention. (**d**) Fluid retention at the conical post-tip of the device. Silhouettes taken by an optical comparator of the profile of a single post suspended above a surface (left) and in contact with a surface (right) show a fluid droplet interacting with the device. Upon device insertion, the gel solution fills the space under the conical post tip, forming the toroid gel. (**e–h**) Schematic of HiExM gel deposition and expansion workflow. (**e**) The device is immersed in a shallow reservoir of gel solution. (**f**) Upon removal, the tip of each device post retains a small volume of gel solution. (**g**) Gel solution is deposited by the device into the well centers of the cell culture plate. Brightfield image (right) shows gel geometry and size prior to expansion. Note that gels deposited in HiExM cover ~1.1 mm$^2$ of the well surface to accommodate the expanded gel, and do not include cells outside the gel footprint. (**h**) Polymerization and expansion are performed with the device in place. Brightfield image (right) shows gel geometry and size after expansion.

The online version of this article includes the following video and figure supplement(s) for figure 1:

**Figure supplement 1.** Schematic depiction of high-throughput expansion microscopy (HiExM) device with displayed features.

**Figure supplement 2.** Overall schematic of high-throughput expansion microscopy (HiExM) comparing standard chemistry and photoinitiation.

**Figure supplement 3.** Detailed workflow of the high-throughput expansion microscopy (HiExM) protocol using photoinitiation.

**Figure supplement 4.** CF conjugated antibodies yield robust signal and resistance to photobleaching in Irgacure high-throughput expansion Microscopy microscopy (HiExM).

**Figure supplement 5.** Acryloyl-X (AcX) and ProteinaseK titration.

**Figure supplement 6.** Residual hoechst signal occasionally remains underneath high-throughput expansion microscopy (HiExM) samples.

**Figure supplement 7.** Schematic depiction of the 24-well plate high-throughput expansion microscopy (HiExM) device.

**Figure 1—animation 1.** Example of expansion process in high-throughput expansion microscopy (HiExM) with ammonium persulfate (APS)/tetramethylethylenediamine(TEMED) chemistry.

**Figure 1—animation 2.** Example of expansion process in high-throughput expansion microscopy (HiExM) with Irgacure chemistry.

reaction. Although we observed gel formation across wells, polymerization and expansion were inconsistent, likely due to inconsistent radical formation and rapid evaporation of the small volume relative to the rate of polymerization (*Figure 1—animation 1*). To overcome this limitation, we tested photochemical initiators including Lithium phenyl(2,4,6-trimethylbenzoyl)phosphinate (LAP), diphenyl(2,4,6-trimethylbenzoyl)phosphine oxide (TPO), and 1-[4-(2-Hydroxyethoxy)-phenyl]–2-hydroxy-2-methyl-1-propane-1-one (Irgacure 2959) which control polymerization of polyacrylamide gels when exposed to UV light (*Li et al., 2022*; *Pawar et al., 2016*). LAP has shown success in slide-based ExM using PEG-based hydrogels (*Blatchley et al., 2022*; *Günay et al., 2023*). We found that Irgacure 2959 allowed reproducible gel formation and expansion when compared to either APS/TEMED or other photoinitiators (*Figure 1—animation 2*).

We next tested the ability of the HiExM platform to achieve nanoscale image resolution of biological specimens. Briefly, A549 cells were cultured, fixed, permeabilized, and immunostained with alpha-tubulin antibodies to visualize microtubules as a reference structure before and after expansion. Stained samples were then incubated with Acryloyl-X (AcX) overnight at 4 °C to allow anchoring of native proteins to the polymer matrix during gel formation. Device posts were immersed in the expansion gel solution containing 0.1% Irgacure 2959. Devices were inserted into the well plate followed by exposure to UV light (365 nm) in an anoxic environment at room temperature to initiate the polymerization reaction. Cells embedded in the resulting gels were digested with Proteinase K then expanded in deionized water overnight (*Figure 1—figure supplements 2 and 3*). Visual inspection using an epifluorescence microscope showed both robust gel formation and fluorescence signal retention across wells. Although we observed bleaching of AlexaFluor dyes in our Irgacure 2959 photopolymerized gels, Cyanine-based Fluorescent (CF) dyes are robust against bleaching in HiExM and result in reproducible signal retention (*Figure 1—figure supplement 4*; *Jiang et al., 2021*). We also titrated AcX and ProteinaseK to optimize signal retention in HiExM and found the 50 µg/mL AcX and 1 U/mL ProteinaseK were most consistent for A549 cells (*Figure 1—figure supplement 5*). This optimization step is likely critical for different cell types. In some cases, we observed residual Hoechst signal underneath expanded cells due to digested cells that were incompletely removed during wash steps during the expansion process (*Figure 1—figure supplement 6*). This signal does not impact the interpretation of results.

A major workflow bottleneck associated with super-resolution imaging, including current ExM protocols, is the significant time and expertise required for image acquisition. For example, the vertical component of expansion increases the depth of the sample, necessitating more optical z-slices to image the full depth of the sample. Moreover, expansion effectively decreases the field-of-view due to increasing sample size, necessitating the acquisition of more images to analyze the same number of cells at lower resolution. To enable rapid 3D image capture of expanded cells in 96-well plates, we used a high-content confocal microscope (Opera Phenix system, Perkin Elmer) (*Figure 2a*). This approach allowed autonomous capture of images across the entire plate at a resolution of ~115 nm compared to ~463 nm for unexpanded samples using the same microscope, objective (63 X, 1.15NA water immersion), and imaging wavelength (*Figure 2—figure supplement 1*). Image resolution was calculated using a decorrelation analysis (*Descloux et al., 2019*). We used the PreciScan plug-in in the Harmony analysis software to first accurately target regions of interest based on Hoechst staining to reduce acquisition times at higher magnification. Briefly, we imaged plates at 5 x across entire wells and stored the coordinates of nuclei for each well. These coordinates were then used to direct the 63 x magnification objective to the relevant fields of view.

We next benchmarked the performance of our platform relative to slide-based proExM (*Tillberg et al., 2016*). Image distortion due to expansion was measured using a non-rigid registration (NRR) algorithm (*Chen et al., 2015*) Prior to the application of HiExM, images of microtubule stained A549 cells were acquired using the Opera Phenix from 61 fields around the center of each well (*Figure 2a*). Cells were then expanded as above and 12 fields were imaged per well. Images from pre- and post-expansion were stitched together and the images for each well were then compared side-by-side to identify matched cells. Although we observed that gels did not maintain a fixed position pre- and post- expansion, close inspection of the images allowed us to align their orientations. Distortion was calculated using the root mean squared error (RMSE) of a two-dimensional deformation vector field by comparing microtubule morphology in the same cell before and after expansion in six to ten fields in each well across six arbitrarily selected wells (*Figure 2c and d*). The average percent error (up to

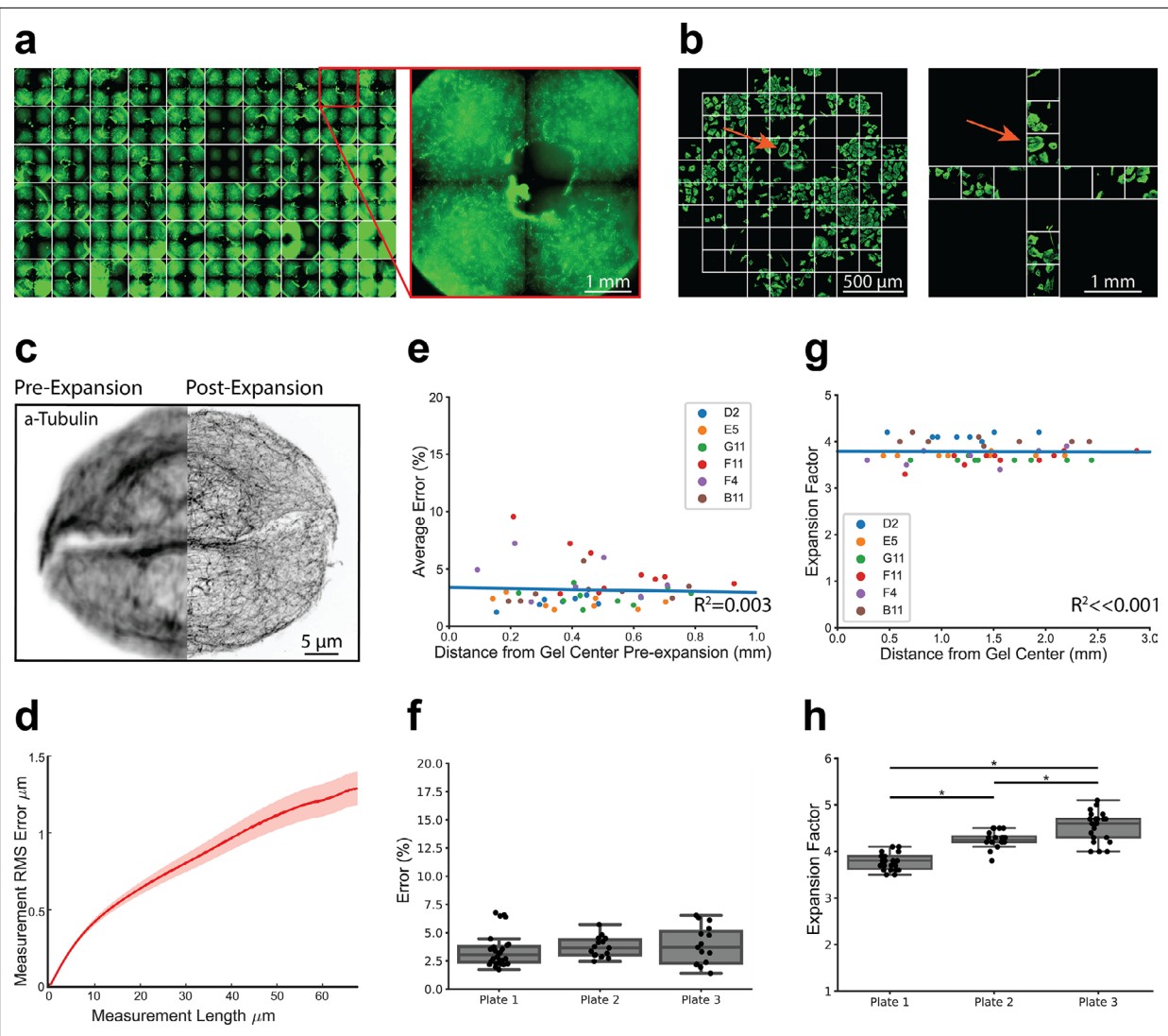

**Figure 2.** High-throughput expansion microscopy (HiExM) is compatible with nanoscale expansion and automated image acquisition of human cells with minimal distortion. (**a**) Expanded gels in a 96-well plate imaged at 5 x with an inset showing an individual gel shown on the right. (**b**) Imaging fields are shown for pre-expansion and post-expansion samples. Prior to expansion, 61 fields were imaged at 63 x around the center of the gel. After expansion, 12 fields were imaged at 20 x magnification. Max projections for each field were then stitched together and manually inspected to identify fields in the pre-expansion and post-expansion image sets containing matching cells. Arrows show an example of a field representing the same cells before and after expansion. (**c**) Representative registered fluorescence image of microtubules before (left) and after (right) expansion showing the increase in resolution conferred by expansion. In this case, both pre-expansion and post-expansion images were taken at 63 x magnification. (**d**) Representative error curve calculated using non-rigid registration of 43 independent fields of view. The shaded region denotes one standard error of the mean. (**e**) Error is not dependent on the location of a given field of view within the gel. Each data point represents the average percent error (up to 40 μm measurement length post-expansion) for a given field of view as determined by an non-rigid registration (NRR) analysis. Seven to ten fields of view were analyzed across six wells. The distance from the center of each field of view to the center of the well was measured manually in FIJI using the stitched pre-expansion image set. (**f**) Box plots showing the distribution of average error among gels across three plates (average = 3.63% +/- 1.39). Average error percentages for each well plate: 3.73 +/- 0.879, 3.89 +/- 1.65, and 3.43 +/- 1.44. (**g**) Expansion factor is not dependent on the location of a given field of view within the gel. Images taken before and after the expansion of the same cell or group of cells were used to measure the ratio of length scales in the two images. (**h**) Box plots showing the distribution of expansion factors of 58 gels across three plates. Expansion factors ranged from 3.5 to 5.1 x across gels with an average of 4.16 +/- 0.394. Average expansion factors for each well plate showed tight distribution with some variability across plates: 3.78 +/- 0.171, 4.26 +/- 0.171, and 4.53 +/- 0.321. Comparison of the three well plates was performed using one-way ANOVA and post-hoc Tukey HSD test (*p-value<0.05).

The online version of this article includes the following figure supplement(s) for figure 2:

**Figure supplement 1.** Representative images post-expansion showing nanoscale resolution in high-throughput expansion microscopy (HiExM).

*Figure 2 continued on next page*

*Figure 2 continued*

**Figure supplement 2.** Infrequent higher expansion error in high-throughput expansion microscopy (HiExM) is due to abnormal stretching and tearing of the gel.

**Figure supplement 3.** Comparison of expansion error measurements between standard expansion microscopy (ExM) chemistry and Irgacure 2959 photoinitiation.

**Figure supplement 4.** High-throughput expansion microscopy (HiExM) performs robustly on cells plated at low and high confluency.

40 µm measurement length post-expansion) was plotted against the distance from the center of the gel (*Figure 2e*). Our analysis found no correlation ($R^2$=0.003) between expansion error and distance from the gel center. In fact, we observed only a small number of outliers (>3 s.d.) in any given well (<2 instances per gel across six gels). These outliers are not expected to affect biological interpretation. The outliers, which are closer to the center of the gel, likely occur when the device post is not in full contact with the well surface, resulting in a higher volume of gel at the center that resists gel swelling under the device (*Figure 2—figure supplement 2*). Expansion error was also highly consistent when comparing wells across multiple plates (average = 3.68%), comparable to slide-based ExM (*Figure 2f*; *Chen et al., 2015*; *Tillberg et al., 2016*). We observed only a small number of image fields (7 of 58) with an average percent error above 6% (maximum 6.78%). Moreover, the calculated error showed that photoinitiation resulted in markedly lower spatial errors compared to APS and TEMED polymerization chemistry (*Figure 2—figure supplement 3*). Analyzing cell seeding densities also showed that cell confluency did not affect image distortion as measured by NRR (*Figure 2—figure supplement 4*).

We next measured expansion factors within individual gels, between gels, and across plates by manually measuring the distance between matched pairs of cell landmarks in FIJI before and after expansion. We found that the expansion factor was highly reproducible across different regions of the gel confirming robust expansion using our method (*Figure 2g*). The average expansion factor of gels sampled across three plates was 4.16+/-0.394 (*Figure 2h*), consistent with proExM (*Tillberg et al., 2016*). We next compared the distributions of expansion factors across three well plates by ANOVA and found statistically significant differences between expansion factor distributions. Thus, while uniform within gels, the expansion factor is somewhat variable between gels and plates. We attribute these differences primarily to the small size of the gels, making them vulnerable to the effects of evaporation. This variability should be taken into consideration for studies where absolute length measurements between plates are important for biological interpretation. Collectively, these data show that the HiExM method results in reproducible nanoscale and isotropic expansion of cells within gels, across wells, and plates.

The ability to image cells at increased resolution using HiExM provides an opportunity to analyze nanoscale cellular features for a range of applications including cell phenotyping as well as drug and toxicology studies. As proof of concept, we analyzed the effects of the cardiotoxic chemotherapy agent doxorubicin (dox) on human induced pluripotent stem cell derived cardiomyocytes (hiPSC-CMs) cultured in 96-well plates (*Figure 3a*; *Johnson-Brbor and Dubey, 2023*). Commonly used dox concentrations in in vitro cardiotoxicity studies range between 0.1 to 10 µM, depending on experimental goals (*Maillet et al., 2016*; *Yu et al., 2020*; *Zhao and Zhang, 2017*; *Stefanova et al., 2023*). Although studies largely focus on electrophysiology, structural changes at the cellular level also provide critical information for assessing cardiotoxicity. For example, changes in nuclear morphology as well as DNA damage are associated with cellular stress and apoptosis, which are commonly observed at high dox concentrations (>200 nM) in in vitro studies using conventional microscopy (*Maillet al., 2016*; *Yu et al., 2020*; *Zhao and Zhang, 2017*; *Stefanova et al., 2023*; *Gewirtz, 1999*). In contrast, heart tissue is potentially exposed to lower dox concentrations in clinical treatments (*Johnson-Brbor and Dubey, 2023*; *Chao et al., 2023*; *Greene et al., 1983*) Because HiExM has the sensitivity to detect nanoscale cellular features, we compared nuclei using Hoechst signal intensity in CMs treated with 1 nM, 10 nM, 100 nM, and 1 µM dox after 24 hr compared with DMSO (*Figure 3b*). Although we did not observe a significant change in overall nuclear morphology (*Figure 3—figure supplement 1*), images showed increasing Hoechst density at the nuclear periphery in a dox concentration-dependent manner.

To quantify this effect, we measured the gradient slope of Hoechst intensity around the nuclear periphery. To this end, we masked 529 nuclei from the background in dox-treated and control samples (*Figure 3c*). We selected a single optical slice that intersected the nucleus perpendicularly along the z-axis for our analysis to avoid an out-of-focus signal at the nuclear edge (*Figure 3—figure*

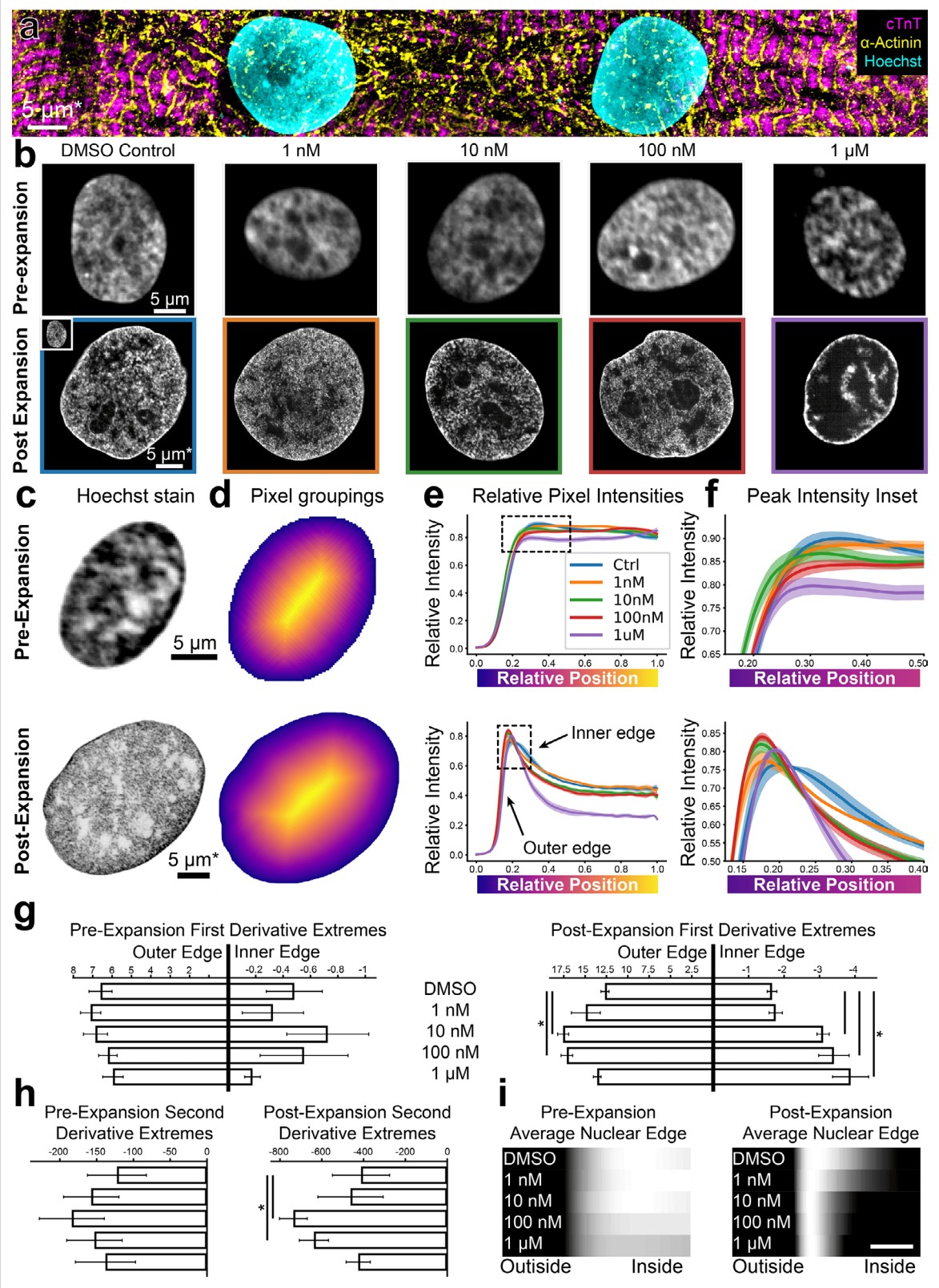

**Figure 3.** High-throughput expansion microscopy (HiExM) shows altered nuclear morphology of cardiomyocytes treated with low doses of doxorubicin. (**a**) hiPSC-CMs imaged post-expansion in HiExM. (**b**) Example images of human induced pluripotent stem cell derived cardiomyocyte (hiPSC-CM) nuclei after dox treatment for 24 hr before (top) and after (bottom) expansion. (**c**) Example images of hiPSC-CM nuclei taken before (top) and after (bottom) expansion. (**d**) Heatmaps of nuclei in (**c**) showing pixel groupings for subsequent analysis. Heatmaps were generated from the image mask after dilation,

*Figure 3 continued on next page*

*Figure 3 continued*

such that the outer edge represents a contour just beyond the nuclear periphery. (**e**) Relative Hoechst intensity plotted as a function of pixel position relative to the edge of the dilated mask in pre- (top) and post- (bottom) expansion nuclei. Colored curves represent dox concentrations or DMSO (ctrl). Shaded regions represent SEM for n=56, 71, 64, 92, and 62 nuclei in the pre-expansion images and n=4 replicates in the expanded case (118, 111, 110, 113, and 77 nuclei analyzed for DMSO control, 1 nM, 10 nM, 100 nM, and 1 µM Dox, respectively). (**f**) Insets of plots in (**e**) highlighting the nuclear periphery. (**g**) Rate-of-change analysis of curves for pre- (left) and post- (right) expansion images. Outer edge values were obtained by determining the maximum values of the derivatives for each condition, and inner edge values were obtained by determining the minimum values of the derivatives within the domain 0–0.4. (**h**) Curvature analysis for pre- (left) and post- (right) expansion data. Values represent the minima of the second derivative for curves within the domain 0–0.4. (**i**) Average Hoechst intensities are plotted as gradients for each dox concentration. Scale bar represents a 10 percent distance from the edge of the nucleus to the center. Error bars represent SEM as in (**e**). * denotes significance from an independent two-sample t-test ($p<0.05$). *Scale bar represents biological scale assuming 4 x expansion.

The online version of this article includes the following video and figure supplement(s) for figure 3:

**Figure supplement 1.** Nuclear volume/area analysis of Doxorubicin-treated cardiomyocytes before and after expansion.

**Figure supplement 2.** Nuclear edge analysis at different optical slices of Doxorubicin-treated cardiomyocytes.

**Figure supplement 3.** Line-scan analysis of Doxorubicin-treated cardiomyocytes.

**Figure supplement 4.** Comparison of derivative curves from *Figure 3* and *Figure 3—figure supplement 3*.

**Figure supplement 5.** Nuclear edge analysis of $H_2O_2$-treated cardiomyocytes.

**Figure supplement 6.** Example of a cell expanded in high-throughput expansion microscopy (HiExM) using Photo-expansion microscopy (ExM) gel chemistry.

**Figure 3—animation 1.** Animation of nuclear periphery analysis employed in *Figure 3*.

*supplement 2*). To account for irregular nuclear shapes, we iteratively grouped pixels into rings for each nucleus starting with the edge of the mask (*Figure 3d*). The mask is dilated outward from the edge of the nucleus to include a background reading as a reference. The average pixel intensity value for each ring represents the average Hoechst intensity as a function of the relative distance from the edge of the nucleus to its center (*Figure 3e*, *Figure 3—animation 1*). Notably, HiExM revealed a peak of Hoechst intensity at the edge of the nucleus that is not resolved in pre-expansion images (*Figure 3f*). A line-scan-based analysis showed the same trend (*Figure 3—figure supplement 3*). To further evaluate these differences, we next calculated the first derivative of each curve and found the maximum and minimum values, which correspond to the gradient slopes on either side of the peak (*Figure 3g*, *Figure 3—figure supplement 4*). In this case, a higher magnitude of gradient slope reflects a 'sharper' transition from the nuclear edge to the background (outside) or from the nuclear edge to the nucleoplasm (inside). Post-expansion, the gradient slopes on both sides of the peak showed a significant increase at dox concentrations as low as 10 nM compared to DMSO, whereas no trend is observed in pre-expansion images (*Figure 3g*).

In addition to gradient slopes at the nuclear edge, we also quantified the pattern of Hoechst signal at the nuclear edge by calculating the curvature of the peak using the second derivative (*Figure 3h*). These data also showed a significant increase in Hoechst intensity at 10 nM dox compared to DMSO, whereas no significant difference is observed using conventional confocal microscopy (*Figure 3i*). We note that, for both the outer gradient slope and peak curvature, values decrease from 100 nM to 1 µM dox, consistent with increased chromatin condensation associated with apoptosis typically observed at higher dox concentrations (*Zhao and Zhang, 2017*; *Toné et al., 2007*). In contrast, CMs treated with increasing concentrations of $H_2O_2$, another cellular stressor, did not show similar differences in Hoechst patterns (*Figure 3—figure supplement 5*). Although direct evidence specifically linking dox to increased DNA condensation at the nuclear periphery is limited, the known pro-apoptotic effects of dox strongly suggest that our observations correlate with these changes (*Maillet et al., 2016*; *Zhao and Zhang, 2017*; *Mobaraki et al., 2017*). Thus, we suggest that HiExM can detect early alterations in response to dox treatment using the Hoechst signal, setting the stage for further investigation to understand the consequences across a range of drugs and conditions. Overall, our analysis demonstrates the ability of HiExM to detect subtle cellular changes compared to standard confocal microscopy that could improve the resolution of drug screening approaches.

## Discussion

Our work demonstrates that HiExM is a simple and inexpensive method that can increase the throughput of expansion microscopy. Another recent method also reports a scalable ExM-based method with comparable throughput, albeit this approach requires the use of a custom cell culture microplate and manual imaging (*Xie et al., 2023*) Our device is fabricated with CNC milling, a common fabrication technique available in most academic settings, making our method widely accessible. The HiExM device can also be produced by injection molding which can further extend its accessibility and distribution in research settings. We have also adapted our device to fit cell culture plates of varying dimensions including a 24-well plate which can accommodate greater expansion and significantly higher resolution (*Figure 1—figure supplement 7*; *Damstra et al., 2022*). Moreover, the use of a photo-active gel solution enabled consistent expansion in our HiExM workflow. In this work, we found that the use of photoinitiation with irgacure caused signal loss for to varying degrees depending on the fluorophore; coupled with the signal dilution inherent to the expansion process, signal intensity is an important consideration when using HiExM. Our platform is amenable to other photo-active expansion chemistries such as the Photo-ExM chemistry which does not require CF dyes or an anoxic environment (*Figure 3—figure supplement 6*; *Blatchley et al., 2022*; *Günay et al., 2023*).

Another advantage of HiExM is that it is fully compatible with autonomous imaging, allowing data acquisition of thousands of cells within a day. HiExM achieved a resolution of ~115 nm in a 96-well culture plate using a 63 x water immersion objective with a 1.15 numerical aperture. By employing microscopes with a higher numerical aperture, we expect our platform can achieve significantly greater resolution. We also show that HiExM error measurements are similar to proExM and suggest that expansion distortions should be considered for robust data interpretation. Recent innovative work that developed a microplate-based ExM protocol maintains hydrogel geometry and orientation improves coordinate-based imaging to accurately assess distortions with ease (*Seehra et al., 2023*). Another recent work used micropatterning to apply expandable protein grids to coverslips for local expansion factor measurement. This method could be adapted to HiExM to enable expansion measurements in large datasets where non-rigid registration is impractical (*Damstra et al., 2023*) Taken together, HiExM is an accessible nanoscale imaging platform designed for scalable sample preparation and autonomous imaging, broadening the use of ExM in biological research and pharmaceutical applications.

## Materials and methods

### Cell culture

Human A549 cells were maintained in culture at a density between $2 \times 10^3$ and $1 \times 10^4$ cells/cm$^2$ in DMEM modified with 10% fetal bovine serum and 1% penicillin-streptomycin. Glass and plastic 96-well plates (Cellvis, P96-1.5H-N or P96-1.5P) were seeded at ~$1 \times 10^4$ and $7 \times 10^4$ cells/cm$^2$ for expansion experiments.

Induced pluripotent stem cell-derived cardiomyocytes were purchased from Cellular Dynamics and cultured on glass-bottom 96-well plates at 30,000 cells/well. For drug treatment, CMs were plated on gelatin for 7 d (Cellular Dynamics) prior to drug treatment and incubated with varying concentrations of Doxorubicin (Tocris 2252) for 24 hr prior to fixation and subsequent sample preparation.

### Immunofluorescence

Briefly, cells were fixed with 4% PFA for 15 minutes at room temperature and then blocked (1% bovine serum albumin and 0.1% Triton X100 in PBS) for 1 hr. Cells were then incubated for 1 hr at 37 °C with primary antibodies in a blocking solution followed by incubation with secondary antibodies in a blocking solution for 1 hr at 37 °C. Cells were washed between each stage in PBS three times at 0-, 5-, and 10 min intervals.

| Primary Antibodies | Vendor | Cat. Number | Concentration |
|---|---|---|---|
| Monoclonal anti-alpha-Tubulin antibody | MilliporeSigma | T5168 | 1:1000 |
| Cardiac troponin T (TNNT2) Rabbit mAb | ABclonal | A4914 | 1:500 |

*Continued on next page*

*Continued*

| Primary Antibodies | Vendor | Cat. Number | Concentration |
|---|---|---|---|
| Monoclonal Anti-α-Actinin (Sarcomeric) antibody | MilliporeSigma | A7811 | 1:500 |
| Secondary Antibodies | Vendor | Cat. Number | |
| CF633 Chicken Anti-Mouse IgG (H+L) | Biotium | 20222 | 1:500 |
| CF568 Donkey Anti-Rabbit IgG (H+L) | Biotium | 20098 | 1:500 |
| CF350 Goat Anti-Mouse IgG (H+L) | Biotium | 20140–1 | 1:500 |
| CF488 Goat Anti-Mouse IgG (H+L) | Biotium | 20010–1 | 1:500 |
| CF543 Goat Anti-Mouse IgG (H+L) | Biotium | 20306–1 | 1:500 |
| CF555 Goat Anti-Mouse IgG (H+L) | Biotium | 20030–1 | 1:500 |
| CF568 Goat Anti-Mouse IgG (H+L) | Biotium | 20100–1 | 1:500 |
| CF633 Goat Anti-Mouse IgG (H+L) | Biotium | 20120–1 | 1:500 |
| Stains | Vendor | Cat. Number | |
| Hoechst 33342 | ThermoFisher Scientific | 62249 | 1:5000 |

## HiExM device fabrication

### 96-well plate HiExM devices

HiExM devices were fabricated from polysulfone (McMaster-Carr 86735K74) using a Trak DPM2 CNC milling machine using the design provided in the supplementary files.

### 24-well plate HiExM devices

24-well plate HiExM devices were fabricated by Proto Labs with injection molding. The device design can be found using the following link:

### Measurement of volume collected by devices

To estimate the volume of liquid collected by the HiExM device, the mass of water was measured on a sartorius TE64 before and after dipping the device into the reservoir. The difference in mass was measured 18 times for an average collected mass of 2806 ug +/-482 ug. The average total mass retained by the device was divided by 12 (the number of posts per device) yielding an estimate of 234 ug or 234 nL per post.

## HiExM workflow

### Irgacure workflow

Cells were fixed with 4% paraformaldehyde (PFA) and immunostained as described above and treated with 50 µM (A549 cells) or 100 µM (cardiomyocytes) Acryloyl-X, SE (6-((Acryloyl)amino)hexanoic acid, succinimidyl ester abbreviated as AcX (Invitrogen A20770)) overnight at 4 °C. Cells were then washed twice with PBS, once with DI water, and the remaining liquid was gently aspirated from the wells. Gel solution composed of 4.04 M sodium acrylate (AK Scientific R624), 0.506 M acrylamide (MilliporeSigma A4058), 0.00934 M n, n'-methylenebisacrylamide (MilliporeSigma M1533), 1.84 M sodium chloride (Invitrogen AM9759) and 1.45 X PBS 1 mL (Invitrogen AM9625) was prepared in a light-protected tube. 1-[4-(2-Hydroxyethoxy)-phenyl]–2-hydroxy-2-methyl-1-propane-1-one (Irgacure 2959, MilliporeSigma 410896) was dissolved in DMSO at a concentration of 1.64 M and this solution was mixed with the gel solution to a final Irgacure 2959 concentration of 0.0328 M at room temperature. The HiExM protocol was performed in a glove bag purged two times with dry nitrogen (MilliporeSigma Z530112). The gel solution was poured onto an absorbent pad at room temperature to form a fluid reservoir. Devices were dipped into the gel solution and immediately inserted into the well plate. The plate was then irradiated for 60 seconds with UVA light using a Uvitron Portaray curing lamp placed 20 inches beneath the plate (Uvitron International UV2270). Approximately 100 µL of Proteinase K (NEB P8107S) digestion solution was added to each gel-containing well and incubated

for ~6 hr at room temperature. After digestion, Hoechst 33324 (Fisher Scientific #62249) in DI water (1:5000 dilution) was added to wells for 5 min, then the 96-well plate was submerged in ~4 liters of DI water overnight under constant agitation while the gels were held in place by the device. Finally, the plate was removed from the water bath, and devices were removed from the well plate for imaging. For each well, excess water was aspirated and two drops of mineral oil was added to secure the expanded gel within the well.

## APS/TEMED workflow

Fixed cells were stained using standard immunofluorescence protocols and treated with anchor solution (6-((Acryloyl)amino)hexanoic acid, abbreviated AcX) (Invitrogen A20770) for at least 3 hr at RT prior to gelation according to standard ExM protocols (*Asano et al., 2018*). Cells were then washed twice with PBS (Invitrogen AM9625) and aspirated dry. Two 2 mL vials of stock X, TEMED (Bio-Rad 1610800), and APS (Millipore Sigma A7460) (40 g/100 mL) were prepared and placed on ice with an aluminum block in a nitrogen glove bag (MilliporeSigma Z530112). The glove bag was purged and backfilled with dry nitrogen twice. 40 uL of APS or TEMED were added to 2 mL of stock X followed by gentle mixing. Both stock X solutions (one containing APS and the other containing TEMED) were poured onto separate absorbent pads. The first device was dipped into the TEMED-containing solution, and stamped into a set of wells, and the process was repeated with the same device to load droplets of TEMED harboring stock X in all wells. Five separate devices were then dipped in the APS-containing solution, then immediately inserted into rows of the well plate. The plate was left on ice for 15 min, then inverted and placed in a glove bag. The underside of the well plate was covered in ~2 mL of water (for thermal conduction) and ~50 °C water bag was laid over the top of the inverted well plate to warm the gels for 5 min. The well plate was removed and 100 uL of digestion solution containing proteinase K (NEB P8107S) was added to each gel-containing well overnight at room temperature. The following day, the well plate was submerged in ~4 liters of DI water under constant agitation for 2 hr, exchanging the water after the first 2 hr. Finally, the plate was removed from the water bath and devices removed from the well plate. Gentle aspiration of the remaining water from the well followed by the addition of ~100 uL of heavy mineral oil over the gel was sufficient to restrict gel movement to allow for automated imaging in a 96-well plate.

## Photo-ExM workflow

Photo-ExM gel solutions were prepared as previously described (*Blatchley et al., 2022*; *Günay et al., 2023*). The protocol for device use with Photo-ExM is the same as the Irgacure protocol with two exceptions. First, gel polymerization in Photo-ExM is not oxygen-inhibited, so the protocol does not require a glove bag. Second, Gel solutions (2 µL) are pipetted into the wells of a custom 3D printed chip (Supplementary File PhotoExM_Gel_Reservoir.SLDPRT) designed for small-volume gel solution storage in HiExM. HiExM devices are applied to the loaded chip such that each post of the device collects gel solution from one of the loaded wells.

## Image acquisition

Confocal imaging was performed on an Opera Phenix imaging system (PerkinElmer). For NRR analysis, pre-expansion images were acquired with a 63 x water objective (NA 1.15) in stacks of six planes spaced 0.5 microns apart and post-expansion images of the same plate were acquired using a 20 x water objective (NA 1.0) in stacks of 50 planes spaced one micron apart. Prior to expansion, 61 tiled fields at the center of each well were imaged at 63 x to capture cells that would be incorporated into the gel. After expansion, 12 fields in a cross pattern were imaged at 20 x to capture a representative set of images for each gel across multiple wells. Based on our analysis, the centers of gels in HiExM should be avoided in imaging as these regions can include cells that are torn and disfigured due to detaching the gels from the device. *Figure 1—animations 1; 2* were obtained on an EVOS M5000 epifluorescence microscope (Thermo Fisher Scientific).

For drug treatment experiments, expanded images were acquired at 63 x magnification using the PreciScan plugin. First, each well was imaged for Hoechst at 5 x in 4 quadrants to cover the full footprint of the well. An analysis in Harmony determined the coordinates of each nucleus in three dimensions and selected a subset of those coordinates to image again at 63 x. The sample sizes for each condition in expanded samples were as follows: 118, 111, 110, 113, and 77 cells for DMSO

control, 1 nM, 10 nM, 100 nM, and 1 µM Doxorubuicn (dox), respectively across the four independent experiments. Pre-expansion images were acquired at 63 x magnification without the PreciScan plugin. Sample sizes for each condition in pre-expansion samples were as follows: 56, 71, 64, 92, and 62 for DMSO control, 1 nM, 10 nM, 100 nM, and 1 µM Doxorubuicn (dox), respectively.

HiExM samples were imaged on the Opera Phenix at 63 x with the following parameters: 100% laser power for all channels; 200 ms exposure for Hoechst, 500–1000+ ms exposure for other channels depending on the strength of the stain and the laser; 60 optical sections with 1- micron spacing; and 4–20 fields of view per well depending on the cell density and sample size requirements. Therefore, imaging one full 96-well plate (60 wells total to avoid outer wells) requires 3 to 64 hr depending on the combination of parameters used.

## Distortion and expansion factor analysis

### Pre-expansion/post-expansion image curation

Prior to applying HiExM, 61 fields of view were imaged at 63 x centered in each well. After expansion, gels were imaged at 20 x in 12 fields of view centered in each well in a cross pattern as in *Figure 2b*. All images were converted to maximum projections. Images from pre-expansion and post-expansion were stitched and examined to find matching cells. For fields of view with matching cells, the corresponding maximum projections for the individual fields of view in pre- and post-expansion were used for subsequent analyses. This image curation is the rate-limiting step in our NRR and expansion factor data analyses due to the time required to identify matched cells.

## Non-rigid registration

Non-rigid registration was performed to measure isotropic expansion within individual gels and across gels. For analysis represented in *Figure 2d*, all fields of view were used in which a cell or group of cells could be identified in both the pre-expansion and the post-expansion image sets. The distance from the center of the analyzed field of view to the center of the well was measured in the pre-expansion image set. For *Figure 2f and a* single field of view near the periphery of the gel was arbitrarily chosen to represent the whole gel. Non-rigid registration was performed using a custom MATLAB analysis pipeline as described (*Chen et al., 2015*). Briefly, pre and post-expansion images are coarsely aligned using the TurboReg plugin for ImageJ (*Jiang et al., 2021*). This pair of outputs is then histogram normalized and masks are generated to exclude regions with no features by applying a Gaussian blur. B-spline-based registration package in MATLAB was used to do non-rigid registration between the images. This procedure was performed on image fields measuring 400px×400px for all analyses.

The resulting data from a non-rigid registration analysis is a curve that represents measurement error as a function of measurement length. To synthesize these curves into single points (as represented in *Figure 2c–f*), The average percent error up to 40 µm measurement length (as measured after expansion) was used. The average percent error is represented as the mean of the ratios of error to measurement length for each individual data point in the error curve up to 40 µm measurement length.

## Expansion factor

To calculate expansion factors, images taken before and after the expansion of the same cell or group of cells were compared. In both images, the distance between two easily identifiable features was measured manually in FIJI, and the ratio of those measurements was used to calculate the expansion factor.

## Nuclear periphery analysis

Image stacks of Hoechst-stained nuclei were compiled and the midplane of each nucleus was manually identified as the plane with the largest nuclear cross-sectional area. Images containing no nucleus or clearly distorted nuclei were omitted from the analysis. A threshold was applied to midplane images to generate masks. These masks as well as their respective raw midplane images were further analyzed using a custom Python script as follows: First, the mask was dilated to encompass the background signal outside of the nucleus. Second, the pixels at the edge of the mask were identified and their position and intensity information were stored. Third, pixels in the 4-neighborhood of each pixel from the outer-edge pixel group were identified and their position and intensity information stored

(note that only pixels inside the boundary defined by the outer-edge pixel group were stored in this way). This third step was iterated for each new 'ring' until all pixels in the dilated mask were grouped. Finally, the mean pixel intensity values of each group were used and the list of mean values was run through linear interpolation to arrive at a list of length 500. To determine the inflection-point slopes of the resulting curves, the first derivative of each curve was calculated using the finite difference approximation. To determine the curvature at the peak intensities of each curve, the second derivative was calculated in the same way. Significance was determined using a two-sample t-test ($p<0.05$).

## Acknowledgements

We thank Paul Tillberg and members of the Boyer and Boyden labs, especially Shiwei Wang, Chi Zhang, and Asmamaw Wassie for helpful discussions. We also thank the Edgerton Student Shop and Mark Belanger for assisting in the fabrication of devices. We thank the Center for the Development of Therapeutics at the Broad Institute and acknowledge funding through the S10 grant NIH OD-026839–01. This material is based upon work supported by the National Science Foundation Graduate Research Fellowship under Grant No. 2141064 and 1122374 to JHD and TS, respectively. This work is supported by NIH EB024261, NIH 1R01AG070831, HHMI, NIH 1R01MH123403, NIH R01MH124606, and NIH 1R56AG069192 to ESB. This research is supported in part by the National Research Foundation, Prime Minister's Office, Singapore under its Campus for Research Excellence and Technological Enterprise (CREATE) programme, through Singapore MIT Alliance for Research and Technology (SMART): Critical Analytics for Manufacturing Personalised-Medicine (CAMP) Inter-Disciplinary Research Group. ESB acknowledges funding from Lisa Yang, Lore McGovern, and John Doerr. This work is also supported by funding from NIH R01HL140471, the Leila and Harold G Mathers Foundation, John J Jarve Seed Fund, and the Deshpande Center to LAB.

## Additional information

### Competing interests

John H Day: John H Day is an inventor on a provisional patent filed for the HiExM device (MIT-064US(02)). Catherine M Della Santina: Catherine M Della Santina is an inventor on a provisional patent filed for the HiExM device. (MIT-064US(02)). Pema Maretich: Pema Maretich is an inventor on a provisional patent filed for the HiExM device.(MIT-064US(02)). Alexander L Auld: Alex L Auld is an inventor on a provisional patent filed for the HiExM device.(MIT-064US(02)). Edward S Boyden: Edward S Boyden is an inventor on a provisional patent filed for the HiExM device.(MIT-064US(02)). Laurie A Boyer: Laurie A Boyer is an inventor on a provisional patent filed for the HiExM device.(MIT-064US(02)). The other authors declare that no competing interests exist.

### Funding

| Funder | Grant reference number | Author |
| --- | --- | --- |
| NIH Office of the Director | OD-026839-01 | John H Day |
| National Science Foundation Graduate Research Fellowship Program | 2141064 | John H Day |
| National Science Foundation Graduate Research Fellowship Program | 1122374 | Tay Shin |
| NIH Office of the Director | EB024261 | Edward S Boyden |
| NIH Office of the Director | 1R01AG070831 | Edward S Boyden |
| NIH Office of the Director | 1R01MH123403 | Edward S Boyden |
| NIH Office of the Director | R01MH124606 | Edward S Boyden |

| Funder | Grant reference number | Author |
|---|---|---|
| NIH Office of the Director | 1R56AG069192 | Edward S Boyden |
| NIH Office of the Director | R01HL140471 | Laurie A Boyer |

The funders had no role in study design, data collection, and interpretation, or the decision to submit the work for publication.

## Author contributions

John H Day, Conceptualization, Data curation, Software, Formal analysis, Validation, Investigation, Visualization, Methodology, Writing – original draft, Writing – review and editing; Catherine M Della Santina, Software, Formal analysis, Visualization, Writing – original draft; Pema Maretich, Alexander L Auld, Conceptualization; Kirsten K Schnieder, Investigation, Methodology; Tay Shin, Resources, Software; Edward S Boyden, Resources, Supervision, Funding acquisition; Laurie A Boyer, Supervision, Funding acquisition, Project administration

## Author ORCIDs

John H Day ⓘ https://orcid.org/0000-0002-5515-6606
Laurie A Boyer ⓘ https://orcid.org/0000-0003-3491-4962

Reviewer #1 (Public review): https://doi.org/10.7554/eLife.96025.4.sa1
Reviewer #2 (Public review): https://doi.org/10.7554/eLife.96025.4.sa2
Reviewer #3 (Public review): https://doi.org/10.7554/eLife.96025.4.sa3
Author response https://doi.org/10.7554/eLife.96025.4.sa4

# Additional files

## Supplementary files

• MDAR checklist

## Data availability

Solidworks files for our devices as well as code for our analysis are available at the following link: https://github.com/lboyerlab/hiExM_Supplementary_Files (copy archived at *lboyerlab, 2024*). Raw data produced from this work are accessible through Dryad (https://doi.org/10.5061/dryad.fbg79cp57).

The following dataset was generated:

| Author(s) | Year | Dataset title | Dataset URL | Database and Identifier |
|---|---|---|---|---|
| John D | 2024 | Data for: High-throughput expansion microscopy enables scalable super-resolution imaging | https://doi.org/10.5061/dryad.fbg79cp57 | Dryad Digital Repository, 10.5061/dryad.fbg79cp57 |

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
