## [Editor Report · eLife Assessment]

This **important** study develops a high throughput version of expansion microscopy that can be performed in 96-well plates. The engineered technology is **convincing** and compatible with standard microplates and automated microscopes and thus will be of broad interest. The application to hiPCS-derived cardiomyocytes treated with doxorubicin provides a **solid** proof-of-concept demonstrating the potential for high-throughput analysis.

---

## [Referee Report · Reviewer #1 (Public review)]

Summary

In this manuscript, Day et al. present a high-throughput version of expansion microscopy to increase the throughput of this well-established super-resolution imaging technique. Through technical innovations in liquid handling with custom-fabricated tools and modifications to how the expandable hydrogels are polymerized, the authors show robust ~4-fold expansion of cultured cells in 96-well plates. They go on to show that HiExM can be used for applications such as drug screens by testing the effect of doxorubicin on human cardiomyocytes. Interestingly, the effects of this drug on changing DNA organization were only detectable by ExM, demonstrating the utility of HiExM for such studies.

Overall, this is a very well-written manuscript presenting an important technical advance that overcomes a major limitation of ExM - throughput. As a method, HiExM appears extremely useful and the data generally support the conclusions.

Strengths

Hi-ExM overcomes a major limitation of ExM by increasing the throughput and reducing the need for manual handling of gels. The authors do an excellent job of explaining each variation introduced to HiExM to make this work and thoroughly characterize the impressive expansion isotropy. The dox experiments are generally well-controlled and the comparison to an alternative stressor (H2O2) significantly strengthens the conclusions.

Weaknesses

(1) It is still unclear to me whether or not cells that do not expand remain in the well given the response to point 1. The authors say the cells are digested and washed away but then say that there is a remaining signal from the unexpanded DNA in some cases. I believe this is still a concern that potential users of the protocol should be aware of.

Editor note: this comment has been addressed in the latest version.

(2) Regarding the response to point 9, I think this information should be included in the manuscript, possibly in the methods. It is important for others to have a sense of how long imaging may take if they were to adopt this method.

Editor note: this comment has been addressed in the latest version.

---

## [Referee Report · Reviewer #2 (Public review)]

Summary:

In the present work, the authors present an engineering solution to sample preparation in 96-well plates for high-throughput super resolution microscopy via Expansion Microscopy. This is not a trivial problem, as the well cannot be filled with the gel, which would prohibit expansion of the gel. They thus engineered a device that can spot a small droplet of hydrogel solution and keep it in place as it polymerises. It occupies only a small portion space at the center of each well, the gel can expand into all directions and imaging and staining can proceed by liquid handling robots and an automated microscope.

Strengths:

In contrast to Reference 8, the authors system is compatible with standard 96 well imaging plates for high-throughput automated microscopy and automated liquid handling for most parts of the protocol. They thus provide a clear path towards high throughput exM and high throughout super resolution microscopy, which is a timely and important goal.

Addition upon revision:

The authors addressed this reviewer's suggestions.

---

## [Referee Report · Reviewer #3 (Public review)]

Summary:

Day et al. introduced high-throughput expansion microscopy (HiExM), a method facilitating the simultaneous adaptation of expansion microscopy for cells cultured in a 96-well plate format. The distinctive features of this method include: (1) the use of a specialized device for delivering a minimal amount (~230 nL) of gel solution to each well of a conventional 96-well plate, and (2) the application of the photochemical initiator, Irgacure 2959, to successfully form and expand toroidal gel within each well.

Addition upon revision:

Overall, the authors have adequately addressed most of the concerns raised. There are a few minor issues that require attention.

Minor comments:

Figure S10: There appears to be a discrepancy in the panel labeling. The current labels are E-H, but it is unclear whether panels A-D exist. Also, this reviewer thought that panels G and H would benefit from statistical testing to strengthen the conclusions. As a general rule for scientific graph presentation, the y-axis of all graphs should start at zero unless there is a compelling reason not to do so.

Editor note: this comment has been addressed in the latest version.

---

## [Author Response]

The following is the authors’ response to the previous reviews.

**Public Reviews:**

**Reviewer #1 (Public review):**
SummaryIn this manuscript, Day et al. present a high-throughput version of expansion microscopy to increase the throughput of this well-established super-resolution imaging technique. Through technical innovations in liquid handling with custom-fabricated tools and modifications to how the expandable hydrogels are polymerized, the authors show robust ~4-fold expansion of cultured cells in 96-well plates. They go on to show that HiExM can be used for applications such as drug screens by testing the effect of doxorubicin on human cardiomyocytes. Interestingly, the effects of this drug on changing DNA organization were only detectable by ExM, demonstrating the utility of HiExM for such studies.Overall, this is a very well-written manuscript presenting an important technical advance that overcomes a major limitation of ExM - throughput. As a method, HiExM appears extremely useful and the data generally support the conclusions.StrengthsHi-ExM overcomes a major limitation of ExM by increasing the throughput and reducing the need for manual handling of gels. The authors do an excellent job of explaining each variation introduced to HiExM to make this work and thoroughly characterize the impressive expansion isotropy. The dox experiments are generally well-controlled and the comparison to an alternative stressor (H2O2) significantly strengthens the conclusions.Weaknesses(1) It is still unclear to me whether or not cells that do not expand remain in the well given the response to point 1. The authors say the cells are digested and washed away but then say that there is a remaining signal from the unexpanded DNA in some cases. I believe this is still a concern that potential users of the protocol should be aware of.

Although ProteinaseK digestion removes most of the unexpanded cells, DNA can sometimes persist. As such, we occasionally observe Hoechst signal underneath cells. The residual DNA is easily differentiated from nuclear Hoechst signal and does not confound interpretation of results. We have added a new supplementary figure that further clarifies this point.

(2) Regarding the response to point 9, I think this information should be included in the manuscript, possibly in the methods. It is important for others to have a sense of how long imaging may take if they were to adopt this method.

We have added detailed information to the methods section to address this point as shown below. In general, we image HiExM samples on the Opera Phenix at 63x with the following parameters: 100% laser power for all channels; 200 ms exposure for Hoechst, 500-1000+ ms exposure for immunostained channels depending on the strength of the stain and the laser; 60 optical sections with 1 micron spacing; and 4-20 fields of view per well depending on the cell density and sample size requirements. Therefore, imaging one full 96-well plate (60 wells total as we avoid the outer wells) takes anywhere from 3 hr to 64 hr depending on the combination of parameters used.

**Reviewer #2 (Public review):**
Summary:In the present work, the authors present an engineering solution to sample preparation in 96-well plates for high-throughput super resolution microscopy via Expansion Microscopy. This is not a trivial problem, as the well cannot be filled with the gel, which would prohibit expansion of the gel. They thus engineered a device that can spot a small droplet of hydrogel solution and keep it in place as it polymerises. It occupies only a small portion space at the center of each well, the gel can expand into all directions and imaging and staining can proceed by liquid handling robots and an automated microscope.Strengths:In contrast to Reference 8, the authors system is compatible with standard 96 well imaging plates for high-throughput automated microscopy and automated liquid handling for most parts of the protocol. They thus provide a clear path towards high throughput exM and high throughout super resolution microscopy, which is a timely and important goal.Addition upon revision:The authors addressed this reviewer's suggestions.
**Reviewer #3 (Public review):**
Summary:Day et al. introduced high-throughput expansion microscopy (HiExM), a method facilitating the simultaneous adaptation of expansion microscopy for cells cultured in a 96-well plate format. The distinctive features of this method include: (1) the use of a specialized device for delivering a minimal amount (~230 nL) of gel solution to each well of a conventional 96-well plate, and (2) the application of the photochemical initiator, Irgacure 2959, to successfully form and expand toroidal gel within each well.Addition upon revision:Overall, the authors have adequately addressed most of the concerns raised. There are a few minor issues that require attention.Minor comments:Figure S10: There appears to be a discrepancy in the panel labeling. The current labels are EH, but it is unclear whether panels A-D exist. Also, this reviewer thought that panels G and H would benefit from statistical testing to strengthen the conclusions. As a general rule for scientific graph presentation, the y-axis of all graphs should start at zero unless there is a compelling reason not to do so.

We have revised Figure S10 to address your comments.